# Luffa Pretreated by Plasma Oxidation and Acidity to Be Used as Cellulose Films

**DOI:** 10.3390/polym11010037

**Published:** 2018-12-27

**Authors:** Ying Wang, Zhao-xuan Ding, Yan-hui Zhang, Chun-yan Wei, Zi-chang Xie

**Affiliations:** 1School of Textile and Material Engineering, Dalian Polytechnic University, Dalian 116034, China; dingzhaoxuan2@126.com (Z.-x.D.); weicy@dlpu.edu.cn (C.-y.W.); xie_zc1225@126.com (Z.-c.X.); 2Shanghai Textile Research Institute Co., Ltd., Shanghai 200082, China; zhangyanhui7712@163.com

**Keywords:** luffa cellulose, dissolution, film, plasma treatment

## Abstract

Cellulose is the most abundant natural polymer on earth. With the increasing shortage of oil resources, people have been focusing more on producing natural cellulose. In this study, guaiacol was used as the model compound to investigate the degradation of lignin in luffa. A new cellulose material was extracted from natural luffa by a pretreatment based on the oxidation and acidity of glow discharge plasma in water. The produced luffa cellulose was dissolved in anhydrous phosphoric/polyphosphoric acid (aPPAC) solvent to prepare cellulose film. Results showed that the reactive species of OH·, HO_2_· and H_3_O^+^ were produced during the plasma discharge of water. The free radicals ·OH were useful in eliminating lignin by the destruction of aromatic structure, whereas H_3_O^+^ in eliminating hemicellulose in the luffa raw material. At the conditions of luffa powder concentration of 9.26 g/L, discharge time of 20 min, and plasma power of 100W, the cellulose component was increased to 81.2%. After 25 min, the luffa cellulose was completely dissolved in the aPPAC solvent at 0–5 °C. Thus, a regenerated cellulose film of cellulose II was prepared. The aPPAC solvent was a good non-derivatizing solvent for the luffa cellulose. The regenerated film exhibited good mechanical properties, wettability and a compact structure. Therefore, plasma pretreatment was an environmentally friendly and high-efficiency method for luffa degumming. The luffa cellulose can be well used in dissolution and regeneration in films.

## 1. Introduction

Cellulose is the most abundant natural polymer on earth. With the increasing shortage of oil resources, people focus more on developing natural cellulose existing in plants, minerals, sea elements, etc. Over the last few years, a number of researchers were involved in investigating the exploitation of natural cellulose from plant fibers due to their affordability and recyclability. Some of these plant fibers have been extensively investigated and widely used as materials in textile or chemical fiber industries (e.g., bamboo and hemp). However, many fibers with very high potential cellulose components (e.g., piassava, caroa, and luffa (sponge gourd), among many others) have not been fully explored yet.

Luffa is a vascular bundle of dried ripe luffa fruits that have a three-dimensional fiber layer and a unique porous physical structure [1]. Nowadays, it is widely employed as filter and adsorption material for dye and metal ions [2,3]. For other industrial applications, luffa is also used as a matrix in reinforcing composite materials. Meanwhile, luffa powder was used for medical purposes to improve skin quality and decrease blood lipid level in ancient China. According to the *Compendium of Materia Medica* records, the juice of luffa was used as a natural remedy for jaundice. Luffa has also been utilized as a bathroom or kitchen accessory after being processed. However, aside from these reports, few previous studies were conducted on luffa cellulose for regeneration purposes in producing films.

The typical mass percentages of natural luffa are 60–65% cellulose, 15–20% hemi-cellulose, and 10–15% lignin. Natural luffa fibers can be regarded as naturally occurring composites consisting mainly of cellulose fibrils embedded in lignin matrix. Apart from rigidity, lignin also provides maximum tensile and strength. However, one of the drawbacks on the applications of luffa is the serious adhesion between lignin and cellulose. The chemical method is effective, particularly in the case of lignin removal by using an alkali hydrogen peroxide solution, and it is widely used in the luffa degumming process [4]. However, this method has many disadvantages, including environmental pollution. Consequently, given increasing green awareness and concern for the environment, an eco-friendly degumming method is being investigated.

The use of low-temperature atmospheric pressure plasma in biology, health care, and medicine elicits an increasing interest and has become one of the main topics of plasma research [5,6]. Many experimental studies have been conducted on the production of a large variety of plasma species, such as reactive oxygen species (e.g., O, OH·, HO_2_·, and O_3_), which are probably the most important species used for cellulose extraction [7]. By controlling the plasma discharge condition, a large variety of hydroxyl free radical ·OH and H^+^ was produced. Free radical ·OH and HO_2_· are useful in eliminating lignin, whereas H^+^ in eliminates hemicelluloses, pectin, and other non associated substances in the luffa raw material [8]. Therefore, the natural luffa in this study was pretreated by glow discharge plasma in water. A new cellulose material was extracted from a natural luffa. The crystal nature of naturally occurring luffa is known as cellulose I. The final extracted cellulose is easily hydrolyzed by acid to water-soluble sugars. Luffa cellulose derivatives, such as cellulose ether and cellulose ester, are the widely used additives in drugs, architecture, food, petrochemical industry and the like. The low cost and reasonable performance of luffa cellulose, including being antibacterial and biodegradable, can fulfill some of the urgent economic and environment protection interests of the chemical fiber and cellulose film industry. 

Solvent plays an important role in the dissolution and regeneration properties of luffa cellulose due to its final application. Many processing methods, such as viscose, *n*-methylmorpholine-*n*-oxid (NMMO) and ionic liquid process, are available to dissolve and regenerate cellulose. The viscose process, using sodium hydroxide as solvent, leads to acritical environmental pollution problem, whereas the NMMO process [9] is very costly. Ionic liquid is a good solvent for cellulose. The dissolution and regeneration in films of luffa in 1-butyl-3-methylimidazolium chloride has been investigated in our previous study [10]. Then in this study, a phosphoric/polyphosphoric acid (PPAC) solvent was prepared by mixing phosphoric acid (PA) and polyphosphoric acid (PPA). The cellulose has been proven to immediately dissolve in the anhydrous PPAC (aPPAC) solvent [11]. It has the following advantages: simple operation, low cost, less time-consuming, and with a low dissolution temperature. Thereafter, the extracted luffa cellulose was dissolved in the aPPAC solvent. The dissolvent process of luffa cellulose in the compound solvent was observed via a polarized optical microscope. The regenerated cellulose film was prepared to maximize the potential applications of natural luffa in cellulose industry. The structure and properties of the luffa regenerated cellulose films were characterized via a scanning electron microscopy (SEM), Fourier transform infrared spectroscopy (FTIR), X-ray diffraction (XRD), mechanical properties, and contact angle measurements. The results were discussed in detail in this paper.

## 2. Materials and Methods

### 2.1. Materials

The dried luffa was purchased from Luohe Commodity Co., Ltd., Luohe, China. The PA, PPA, crystal violet (CV), *n*-butanol and guaiacol were obtained from Tianjin Kemiou Chemical Reagent Co., Ltd., Tianjing, China.

### 2.2. Plasma Treatment

The CTP1200 plasma treatment equipment was made by Coronalab, Nangjing, China. The reactor is shown in Figure 1. The dielectric barriers of this reactor were quartz glass, with a polar distance of 8mm. The aqueous solutions (e.g., luffa and guaiacol) were placed into a plasma reactor.

### 2.3. Evidence of the Radical Produced by Plasma

To gather the evidence of free radicals produced by plasma in water, the experiment was performed as follows (as listed in Table 1). Before the plasma discharge, the distilled water was placed on the surface of the quartz electrode and treated in the discharge power of 100W, polar distance of 8mm, and treatment time of 30 s. After the treatment, the water was quickly removed into a 0.1% CV solution. The CV dye is a radical indicator. Then, a radical scavenger, *n*-butanol, was added. The absorbance of CV solutions was measured using the UV-8000 (Shimadzu Corporation, Tokyo, Japan) ultraviolet visible (UV) spectrophotometer, and the absorbance was obtained.

### 2.4. Drawing the Standard Curve of Guaiacol

To obtain the degradation condition of lignin, the guaiacol was used as the model compound. Different concentration of guaiacol-water solutions was prepared, and the absorbance of guaiacol solutions was determined using the UV spectrophotometer in the range of 200–500 nm. The standard curve of the guaiacol was drawn with the guaiacol concentration as the abscissa and maximum absorbance as the ordinate.

### 2.5. Experiments of the Guaiacol and Luffa Degradation

10 mL of 0.75 g/L guaiacol-water solution was placed into the plasma reactor. The guaiacol solution was treated under the discharge powder of 100W. 100W was the best discharge power that can be judged by naked eyes. The guaiacol solutions were treated under 5 min, 10 min, 15 min and 20 min, respectively. The untreated and plasma-treated guaiacol solutions were diluted during the UV absorbance measurement. The degradation rate *C* of guaiacol is calculated using Equation (1).
(1)$$C=\frac{C_{0}-C_{e}}{C_{e}}\times 100\%$$ where *C*_0_ is the concentration of guaiacol solution and *C*_e_ is the concentration of guaiacol after plasma treatment, which was calculated according to the standard curve of the guaiacol.

The experiment of luffa degradation was the same as the process of guaiacol solution. Before the experiment, the luffa is grinded into powders.

### 2.6. Preparation of the aPPAC Solvent

The PPAC solvent composition is defined by the P_2_O_5_ content. PA and PPA correspond to P_2_O_5_ concentration of 72.4 wt % and 84.0 wt %, respectively. In this experiment, the PPAC solvent was prepared by mixing PA and PPA. The prepared aPPAC solvent was stirred at 30 °C for 120 min. The mixed PA and PPA solvent had the anhydrous state at 74 wt % of P_2_O_5_ through the anhydrous CuSO_4_ powder testing. Then, the aPPAC solvent of 74 wt % of P_2_O_5_ was prepared.

### 2.7. Polarized Optical Microscope Observation

In observing the dissolution process of the luffa cellulose in the aPPAC solvent, the HPL-85 polarized optical microscope (Shanghai Quantong Optical Co., Ltd., Shanghai, China) was used to trace the dissolution process. The applicable environmental temperature of the HPL-85 is within 5–35 °C. Before the observation, the aPPAC solvent was precooled to 0 °C, and the environmental temperature was adjusted to 5 °C. Then, some extracted luffa cellulose and several drops of the aPPAC solvent were placed on the sample stage. The observation time was from 2 min to 30 min, and the magnification of the microscope was 100. 

### 2.8. Preparation of the Regenerated Cellulose Films

The pretreated luffa cellulose was added into the aPPAC solvent (0 °C) at concentrations of 5%, 10%, 15%, and 20%. The luffa cellulose-aPPAC solution was stirred using a magnet stirrer to make it to a homogeneous solution. The stirring time was determined by the results of the polarized optical microscope observation.

The produced luffa cellulose-aPPAC homogeneous solution was placed on a clean glass slide and spread by a smooth glass rod. After spreading the film, the glass slide was quickly immersed into ice water for the 30 min coagulation. Thereafter, the luffa cellulose films were washed two times to remove the residual solvent, dried at room temperature, and stored in a silica gel dryer. The regeneration of the cellulose film occurred during the coagulation of the film.

### 2.9. Characterizations

The surface morphology of the luffa and luffa cellulose film was observed via a SEM. The samples were coated with gold and examined using the JEOL JSM-6460LV microscope (Tokyo, Japan). The crystalline structure of the untreated luffa and luffa cellulose film was examined using a wide-angle X-ray diffractometer (D/max-3B; Rigaku Co., Ltd., Tokyo, Japan). The samples were scanned from 2θ = 10–50° with an operating voltage and current of 40 kV and 200 mA, respectively. The radiation used was Ni-filtered kα radiation with a wavelength of 1.5406 Å. The chemical properties of natural luffa and luffa cellulose film were tested via the FTIR (Spectrum One-B; Perkin Elmer Co., Ltd., Waltham, MA, USA). The FTIR spectra were recorded in a scanning range of 400–4000 cm^−1^ at a resolution of 4 cm^−1^. The scan speed was 0.2 cm/s.

The chemical component analysis of the luffa was conducted using the TZ/T 30001-92 “analysis method of ramie fiber chemical composition”. The mechanical properties of the regenerated cellulose films were measured using the TH-8102S material testing machine (Suzhou Tuobo Machinery Equipment Co., Ltd., Suzhou, China). The samples were cut into 50 mm × 10 mm [12], the clamping length was 30 mm, and the stretching speed was 10 mm/min. Each sample was measured three times to obtain the average value. The polymerization degrees of the cellulose were measured using the copper-ethylenediamine method. The hydrophilicity of the luffa cellulose film was tested using a contact angle meter (K100; Kruss Co., Ltd., Hamburg, Germany).

## 3. Results and Discussion

### 3.1. Evidence of the Radical Produced by Plasma

In the dielectric barrier discharge, the energetic electrons can collide with the background molecules (e.g., N_2_, O_2_, and H_2_O) producing secondary electrons, photons, ions, and radicals. Reactive molecular dynamics simulations have been showed that oxygen species of O atoms, HO_2_· radicals, OH· radicals, and H_2_O_2_ produced during the energetic electrons colliding with water [7,8].

CV dye is the indicator of oxidative radicals. The oxidative radicals can oxidize and degrade dyes, lighten their color, and reduce the absorbance value. To prove that the plasma discharge system contains radicals, the CV solution was added into plasma-treated water. Figure 2 shows the spectra of samples (a), (b), and (c). After the reaction of the plasma-treated water with CV, the absorbency peaks of solution (b) remarkably decrease compared with solution (a). This result indicates that oxidative radicals are present in the water produced by the plasma and resulting in the CV degradation. After pouring the *n*-butyl alcohol, sample (c), the maximum absorbency of the CV solution was 44%, slightly higher than sample b. Given that *n*-butyl alcohol is a free radical trapping agent, the absorbency enhancement also proves that the oxidative radicals are produced in the water. The oxidative radicals are likely one of the HO_2_·radicals or OH· radicals or both of them.

In order to determine what type of radicals exist in water, the pH of the plasma treated water was investigated. The experimental results showed that the plasma-treated water exhibited acidity with the pH of 3.73. In the free radicals, HO_2_· and OH·, HO_2_ is a superior proton donor but is a weaker proton acceptor than water [8]. Hence, the result of the HO_2_ reaction can be expressed in Equation (2). This result indicates that HO_2_· radicals are present in the plasma treated water. Noticeably, a large number of hydrated protons are produced in water. Yang [13] did an experiment of plasma precipitation of calcium ions in hard water. It was found that alkalinity of water was weakened, indicating the characteristics of plasma acidity in water environment. Note that in Equation (2), with the death of the HO_2_ free radicals, the reaction will go on in the reverse direction. The concentration of H_3_O^+^ will gradually decrease, which resulted in the finally destruction of the acidic water.
HO_2_· + H_2_O ←→ [O_2_−H-OH_2_] ←→O_2_^−^ + H_3_O^+^(2)

Note that the CV solution or *n*-butyl alcohol was manually poured into the plasma-treated water, thus consuming at least several seconds. Generally, the lifetime of the OH· radical is a few μs in a gaseous state. However, apart from HO_2_· radicals, we detected other radicals in water after the discharge in several seconds in our experiments. It is the OH· radicals. The theoretically simulated calculation also indicates that the interaction of OH· radicals with water leads to the consecutive formation of another OH· radical [8]. The reaction scheme for HO· in water can be expressed in Equation (3). Noticeably, the lifetime of the hydroxyl radicals in water can be a few hundred times greater than that in the gas state.
HO· + H_2_O ←→ [HO-H-OH] ←→H_2_O + HO(3)

Therefore, it is the OH· radicals, produced during plasma water discharge, which attack the dye molecules, resulting in the bleaching of the CV solution. The OH· and HO_2_·radical species are evidently produced during the plasma discharge. HO· can survive longer than the other reactive species in water solution. HO_2_·reacts with water transformed into H_3_O^+^, thereby enabling the plasma-treated water to produce acidity.

### 3.2. Experiment of the Guaiacol Degradation

Figure 3 is the UV absorbance curves of the guaiacol-water solutions. For all the guaiacol solutions, there are clear absorption peaks at 274 nm, which is attributed to the aromatic ring in the guaiacol molecule. Table 2 lists the absorbance of guaiacol solutions with different concentration at 274 nm. Then the standard curve of guaiacol was drawn. According to the data in Table 2, the linear fitting result of guaiacol concentration-absorbance is *y* = 0.016*x* + 0.048. The correlation coefficient is 0.997.

### 3.3. Experiment of the Luffa Degradation

Figure 4 is the UV absorbance curves of plasma-treated guaiacol-water solutions. For untreated guaiacol solution, in the curve (a), there is a clear absorption peak at 274 nm of the benzene structural unit in the guaiacol molecular. However, a big change happens in the absorbance curves when the plasma treated time extends to 5 min. A large decreasing of absorption peaks appears at 274 nm. Also, there is a weak absorption peak from 270 nm to 305 nm. This result indicates that the aromatic rings in the guaiacol are destroyed. Many complexes are produced after the pyrolysis of benzene rings. The produced complexes cause the maximum wavelength red shift and decreases the absorbance. With the increasing of plasma treated time from 10 min to 20 min, the absorbance at 274 nm gradually decreases with time. The results show that more benzene rings of guaiacol are destroyed and the guaiacol solution is degraded. Table 3 lists the degradation rate of plasma-treated guaiacol solutions. The degradation rate of guaiacol increases gradually along with the time. After being treated for 20 min, the degradation rate of guaiacol is as high as 80%. 

During the plasma discharge of water, a large number of OH· and HO_2_· radicals are produced, which has been proved by the above experiments. On the one hand, these oxidative radicals will attack benzene rings and form benzene radicals. Benzene radicals can capture other free radicals to produce biphenyl compounds. On the other hand, The C=C of benzene and biphenyl compounds can be further oxidized and destroyed. Many complexes with one or more functional groups, such as carbonyl groups, carboxyl groups, and among many others, are produced. These compounds can continue to be degraded into small molecule such as CO_2_, H_2_O, etc. This can be explained the result of a weak absorption peak at 270–305 nm.

### 3.4. Experiment of the Luffa Degradation

The OH· and H_3_O^+^ reactive species are therefore produced during the plasma discharge of water. The oxidative radicals are useful in eliminating lignin whereas H_3_O^+^ in eliminating hemicellulose, pectin, and other non associated substances in the luffa raw material. Then the degradation of luffa was carried out with a discharge time of 20 min and plasma power of 100 W. Based on the 8.1% lignin content in the luffa fiber presented in Table 4, and the treated guaiacol of 0.75 g/L, the estimated luffa powder concentration was 9.26 g/L. 

After plasma treatment, the cellulose content of luffa fiber increases to 81.2% and the hemicellulose and lignin contents decrease. The results show that the removed rate of lignin is approximately 45%. However, using the chemical method [14], treatment was carried out by steeping the luffa in 2% NaOH for 90 min or 1% methacrylamide aqueous solutions for 180 min. The lignin removal rate of these treatments was 27% and 7%, respectively. Thus, the lignin removal is more effective in this study. 

Figure 4d is the UV absorbance of residue of plasma-treated luffa. There are also nonclear absorption peaks at 274 nm, indicating that the benzene structure of lignin is destroyed. Note that there is an obvious and broad absorption peak at 310 nm. It is very different from the curve of plasma-treated guaiacol solutions. As is well known, the chemical structure of natural lignin is more complex than guaiacol. Particularly, in the gramineous plants, the lignin contains a certain amount of phenyl-coumarin structure. The phenyl-coumarin structure will be dehydrated to phenyl-coumarone at the acidic conditions. Phenyl-coumarone has a clear absorption peak at 310 nm. Then the clear absorption peak of 310 nm is attributed to the phenyl-coumarone of luffa. In addition to the complexes with carbonyl, carboxyl, etc., the complexes with ketone group are produced after the destruction of lignin. Meanwhile, this result gives indirect evidence that plasma-treated water is a kind of acidic water.

In this study, the energy consumption of luffa degradation is 120 kJ per 9.26 g, 13 kJ per gram luffa powder. It is a little higher than the electrical energy consumption of a ramie degumming factory in China. This is a major problem in plasma treatment. The biggest advantage of this method is the post-treatment of degumming residues. It is simple and environmentally friendly for no chemicals to be used in the degumming process. Moreover, as time went by the acidity of the plasma treated water would fade gradually [15]. It is great benefit for the recycling of water and has important significance for energy saving and for decreasing air pollution. The reason for acidic water turning into neutral water has been clarified in Equation (2). Then, it is concluded that plasma treatment is an environmentally friendly and high-efficiency method for luffa degumming. The degumming experiments of other materials, such as hemp and ramie, are now performed.

### 3.5. SEM of the Luffa by Plasma Treatment

Figure 5 presents SEM micrographs of the surface morphology of the untreated and pretreated luffa. As presented in (a) and (c), it can be seen that the fibers show a homogeneous aspect, with a rough surface and an outer lignin rich layer around the fiber. Most of the fibrils of the untreated luffa fibers are glued together, thereby forming a thick fiber with a diameter of approximately 100–150 μm. The internal fibrils are not observed. However, as showed in (b) and (d), the plasma-treated luffa fibers display an irregular surface. The formation of surface indentations can be seen clearly. They are the results of plasma treatment due to a great level of surface material being removed. Figure 5e is the cross section of the untreated luffa. The SEM image shows the complicated physical structure in the natural luffa, which forms a natural, porous mat, similar in structure to that of a sponge. As presented in (f), in the cross section of plasma-treated luffa, a permanent damage of fiber integrity is more evident. There are many pores and gaps in the fractured surface, which consequently increase the exposing opportunity of the fiber inner layers to solvent. Thus, the dissolution of luffa cellulose in the solvent becomes easy and the dissolution rate of luffa cellulose will be enhanced.

It is interesting to note that plasma treatment leaves the bulk single fiber intact and only modifies the outer surface layers. This feature may improve the interfacial phenomena when the obtained luffa fibers are used as absorption materials, filters, and in polymeric composites. Moreover, this is similar to the result that Tanobe et al. [14] had reported for luffa fibers treated by NaOH and methacrylamide. Thus, compared with the chemical pretreatment, it is concluded that plasma treatments greatly reduce the treatment time and do not require the use of solvent.

### 3.6. Polarized Optical Microscope Observation

By using the polarized optical microscope, one can see the molecular orientation of the cellulose. If the cellulose structure disappears in the optical microscope during the dissolution process, then the cellulose has been dissolved completely in the solvent. The images of the pretreated luffa dissolving in the aPPAC solvent obtained from the polarized optical microscope are shown in Figure 6a–d. The figures illustrate the dissolution process of the luffa cellulose in the aPPAC solvent.

A large number of clubbed luffa fibrils appear in the aPPAC solvent in 2 min. Meanwhile, the number of luffa fibrils clearly decrease in 10 min. After 15 min, only a few fibrils are observed. When the dissolution continues up to 25 min, nothing is visible in the optical photograph. Hence, the fibrils are completely dissolved in the aPPAC solvent. The reaction between H_4_PO_4_^+^ and the hydroxyl groups on cellulose chains leads to cellulose dissolution in the aPPAC solvent. Therefore, the dissolution time of the extracted luffa cellulose in the aPPAC solvent was 25 min within the circumstance temperatures of 0–5 °C. Subsequently, the luffa cellulose-aPPAC solution was prepared for stirring for 25 min.

Compared with the NMMO cellulose dissolution system of more than 8–10 h at 80–100 °C [9], the dissolution time of the cellulose in the aPPAC solvent is faster, and the dissolution temperature is lower. Within a temperature of 0–5 °C, only 25 minis required for the cellulose to dissolve.

### 3.7. Preparation of the Luffa Cellulose-aPPAC Film

Based on the polarized optical microscope results, the luffa cellulose-aPPAC solutions were prepared at 25 min and 0 °C, with concentration of 5%, 10%, 15%, and 20%, respectively. For the 5% concentration, a smaller cellulose chain was observed in the solution, resulting in a low viscosity of the solution. For higher concentration of up to 20%, many molecular chains would increase the viscosity of the solution. Preparing a film both in a higher and lower viscosity is difficult. Thus, we only used a 15% solution in this study. The regenerated luffa cellulose film was prepared by casting the luffa cellulose-aPPAC solution (15%, 25 min, and 0 °C) in ice water for 30 min of coagulation.

### 3.8. XRD Analysisof the Luffa Cellulose-aPPAC Film

Figure 7 shows the XRD spectra of the untreated luffa fibers and regenerated luffa cellulose films. The characteristic peaks of the untreated luffa fibers appear at 2θ = 15.7°, 22.4°, and 34.4°, thereby matching with cellulose I. Meanwhile, the main peaks of the regenerated luffa cellulose films are presented at 2θ = 11.9° and 21.5°, thereby matching with cellulose II. Therefore, the original lattice of the luffa is altered under the action of the aPPAC solvent.

### 3.9. FTIR Analysis of the Luffa Cellulose-aPPAC Film

The chemical structure of luffa fibers and regenerated luffa cellulose films was characterized by FTIR. Figure 8 shows the FTIR spectra, and Table 5 specifies the observed absorptions. The untreated luffa in curve a shows the intense, characteristic peaks of the lignocelluloses materials at approximately 3360 cm^−1^ (O–H stretching band), 2900 cm^−1^ (CH stretching band), 1750 cm^−1^ (C–O stretching band), 1646 cm^−1^ (O–H deformation band), 1429 cm^−1^ and 1062 cm^−1^ (characteristic peaks related to the cellulose), and 1500 cm^−1^ and 1469cm^−1^ (characteristic peaks related to lignin).

After the regeneration process shown in curve b, the film shows the strong characteristic peaks related to the cellulose and no new functional groups were produced. One of the alterations in the regeneration process is that the cellulose in film does not show the absorption peaks at 1469 cm^−1^ and 1500 cm^−1^. The disappearance of these two peaks is due to the removal of lignin by the destruction of benzene rings. Obviously, there is a band of C=O at 1750 cm^−1^, which has a lower intensity as that in curve a. Probably, it is the by-product of lignin degradation, complexes with carbonyl, carboxyl, ketone group, and the like. Of course, itis also a possibility for the exposure of inner lignin, or for both of them together.

Evidently, part of lignin has been removed by the plasma pretreatment. No significant differences between the chemical structure of the untreated luffa and luffa cellulose film were found. Moreover, the derivatization reaction does not occur during the regeneration process of the luffa cellulose in films. Removal of all lignin in its native state from the fibrils remains a problem for further investigation. 

### 3.10. SEM Analysis of the Luffa Cellulose-aPPAC Film

As shown in the cross section presented in Figure 9, the surface of the film is not very smooth at 3000× magnification. The roughness surface probably is formed in the coagulation bath. The surface of the film coagulated rapidly. But the inside of the film coagulated slowly. No big cellulose particles appeared and few voids appeared in the cross-section, indicating that the luffa cellulose has completely dissolved in the aPPAC solvent and a compact film is obtained.

### 3.11. Mechanical Properties and Contact Angle Measurement of the Regenerated Cellulose Films

The breaking strength of the regenerated cellulose films was 36.78 MPa, the breaking elongation of the films was 31.55%, and the polymerization degree of the cellulose of the dried luffa was 714. After the treatment, the polymerization degree of the luffa cellulose became 564. The contact angle measurements of the regenerated luffa cellulose films are listed in Table 6. The average contact angle of the cellulose film was 16.38°. The result demonstrates that the films exhibit good mechanical properties and wettability.

## 4. Conclusions

In this study, guaiacol was used as the model compound to investigate the degradation of lignin in luffa. The evidence of the oxidative radical produced by plasma was done. The results showed that the OH· and H_3_O^+^ reactive species were evidently produced during plasma discharge. The free radical ·OH was useful in eliminating lignin by the destruction of aromatic structure, whereas H_3_O^+^ in eliminating hemicellulose in the luffa raw material.

The new cellulose material was extracted from natural luffa by a pretreatment based on the oxidation and acidity of glow discharge plasma in water. With the conditions of powder concentration of 9.26 g/L, discharge time of 20 min, and plasma power of 100 W, the component of the cellulose was increased to 81.2%. Part of lignin had been removed by the plasma pretreatment for the destruction of lignin. The destroyed results of luffa lignin were the complexes with carbonyl, carboxyl, ketone group, and the like.

The luffa cellulose-aPPAC dissolution system was also investigated. The results showed that the aPPAC solvent was a good non-derivatizing solvent for the luffa cellulose, and the anhydrous state of the PPAC was achieved at 74 wt % of P_2_O_5_ concentrations. The luffa cellulose completely dissolved in the aPPAC solvent at 0 °C and 25 min for a 15% solution.

The regenerated luffa cellulose film was successfully prepared by casting the luffa cellulose-aPPAC solution in ice water for 30 min coagulation. After being regenerated in films, the crystalline structures of the luffa cellulose film were transformed from cellulose I to cellulose II. The luffa cellulose film appeared the characteristic functional groups related to the cellulose. No derivatization reaction occurred during the dissolution process. The films also exhibited good mechanical properties, wettability and a compact structure. 

Therefore, a new cellulose material was extracted from natural luffa by a pretreatment based on the oxidation and acidity of glow discharge plasma in water. This was an environmentally friendly and high-efficiency method for luffa degumming. The biggest advantage of this method is the simple and environmentally friendly post-treatment of residues. Moreover, luffa cellulose is a new source suitable in producing regenerated cellulose films. Further investigations on crystallinity and the penetration properties of luffa cellulose films are ongoing.

## Figures and Tables

**Figure 1 polymers-11-00037-f001:**
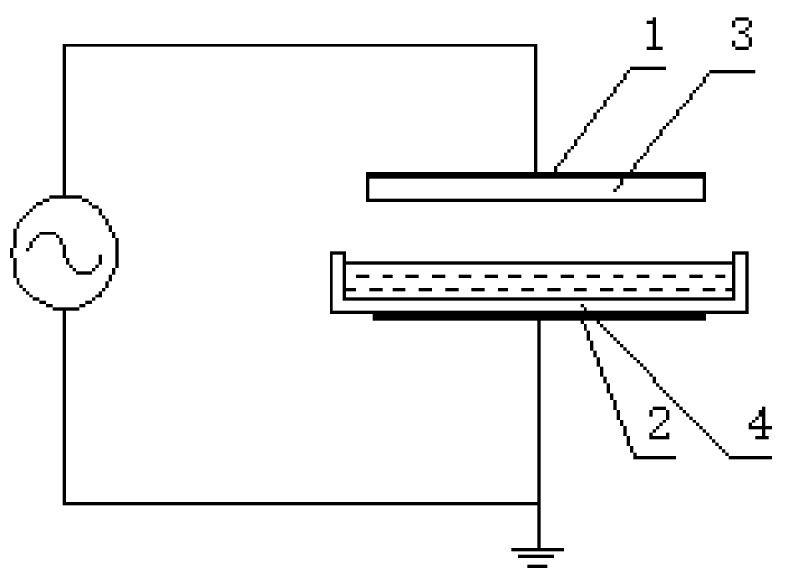
Chart of the plasma reactor: 1and 2 are electrodes and 3 and 4 are barriers.

**Figure 2 polymers-11-00037-f002:**
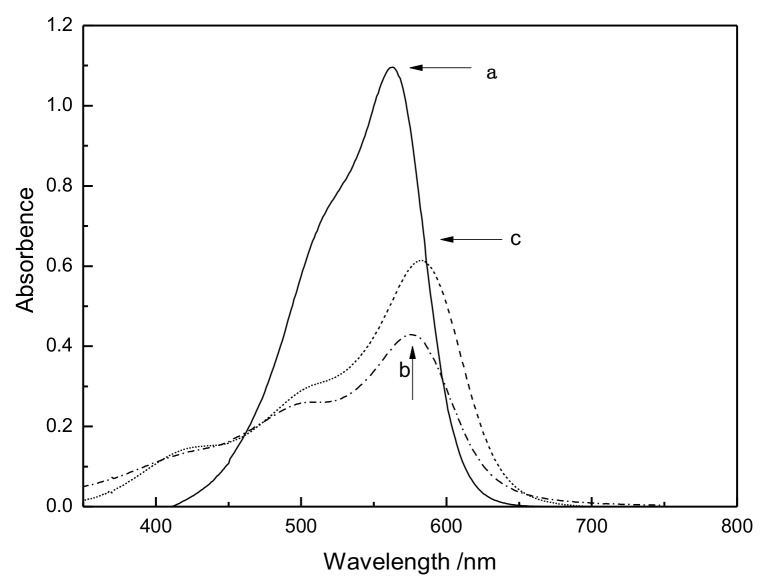
UV spectra of (**a**) CV solutions;(**b**) CV solutions with plasma-treated water and (**c**) CV solutions with plasma-treated water and *n*-butyl alcohol.

**Figure 3 polymers-11-00037-f003:**
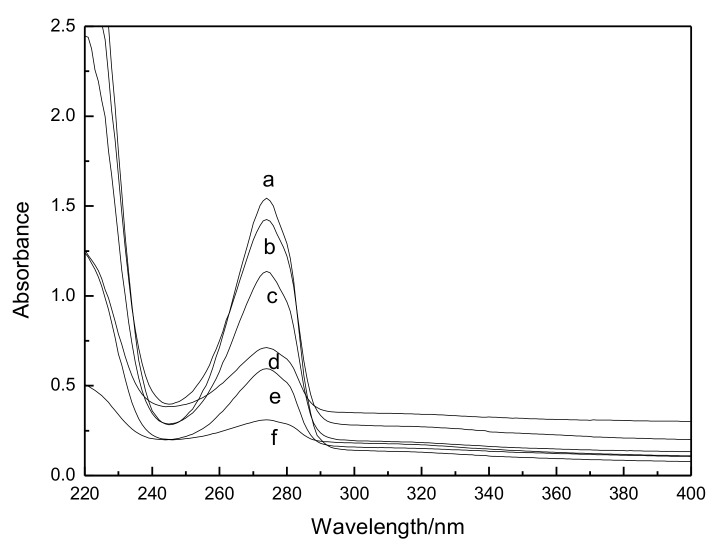
UV absorbance curves of guaiacol-water solutions of (**a**) 95 mg/L; (**b**) 80 mg/L; (**c**) 65 mg/L; (**d**) 40 mg/L; (**e**) 35 mg/L and (**f**) 15 mg/L.

**Figure 4 polymers-11-00037-f004:**
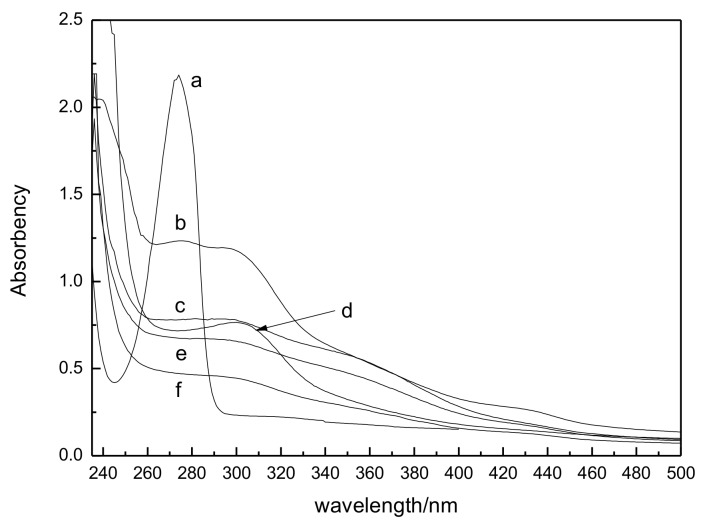
UV absorbance curves of plasma-treated solutions at (**a**) 0 min; (**b**) 5 min; (**c**) 10 min; (**e**) 15 min and (**f**) 20 min and luffa at (**d**) 20 min.

**Figure 5 polymers-11-00037-f005:**
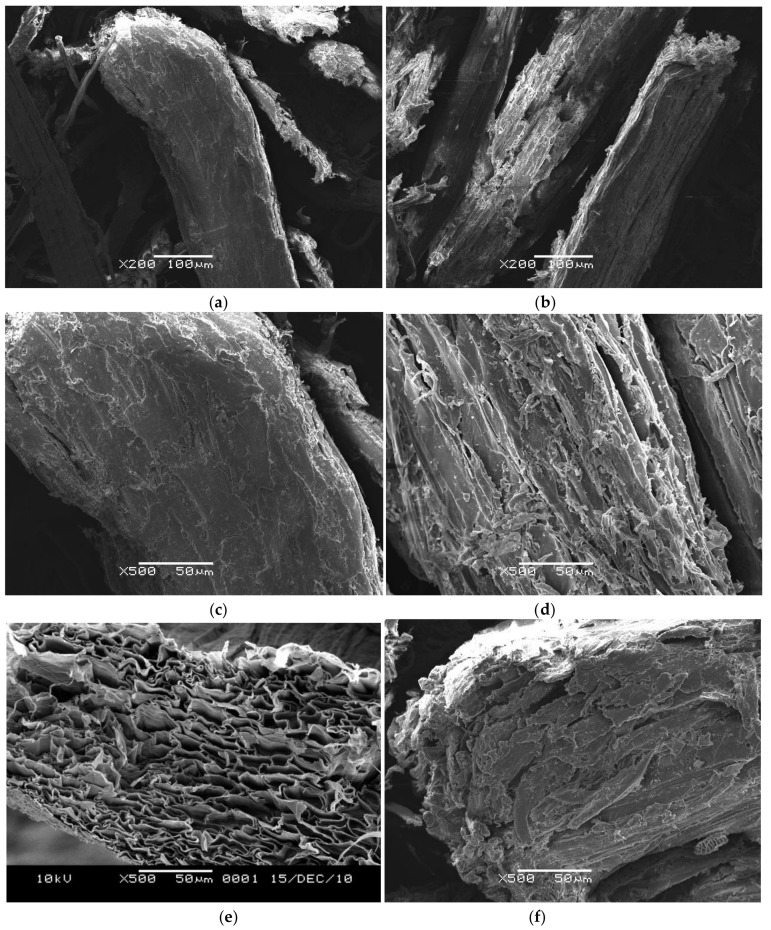
SEM images of (**a**) untreated luffa at 200× magnification; (**b**) plasma-treated luffa at 200× magnification; (**c**)untreated luffa at 500× magnification; (**d**) plasma-treated luffa at 500× magnification; (**e**) cross section of untreated luffa at 500× magnification and (**f**) cross section of plasma-treated luffa at 500× magnification.

**Figure 6 polymers-11-00037-f006:**
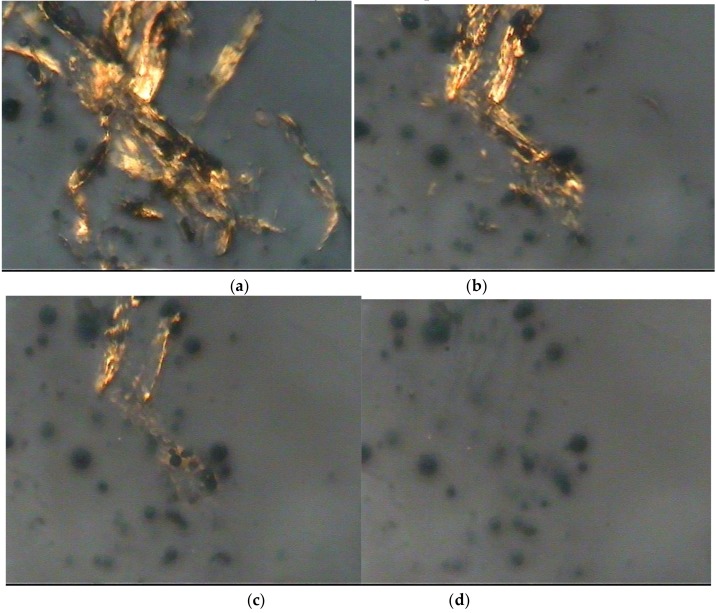
Dissolution process of luffa cellulose in the aPPAC solvent at (**a**) 2 min; (**b**) 10 min; (**c**) 15 min and (**d**) 25 min.

**Figure 7 polymers-11-00037-f007:**
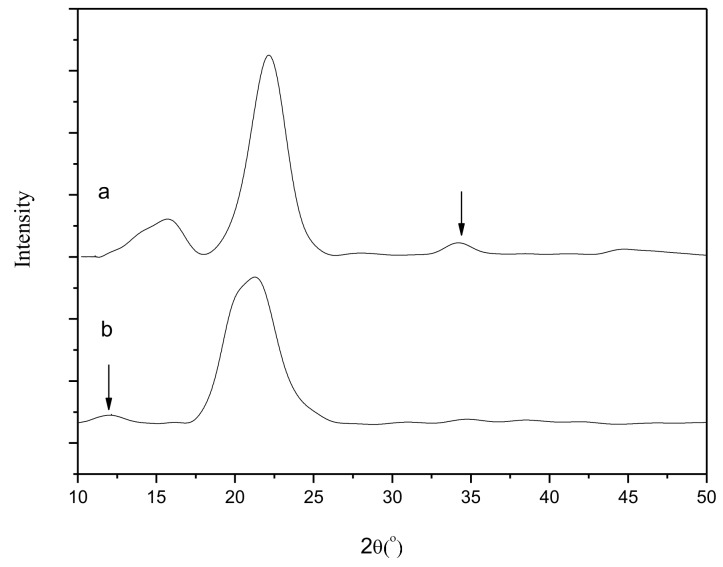
XRD spectra of (**a**) untreated luffa and (**b**) regenerated luffa cellulose film.

**Figure 8 polymers-11-00037-f008:**
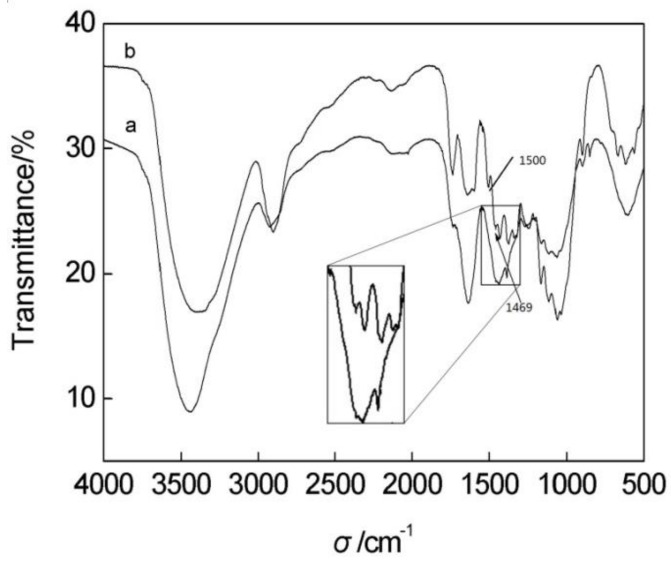
FTIR spectra of (**a**) regenerated luffa cellulose film and (**b**) untreated luffa fiber.

**Figure 9 polymers-11-00037-f009:**
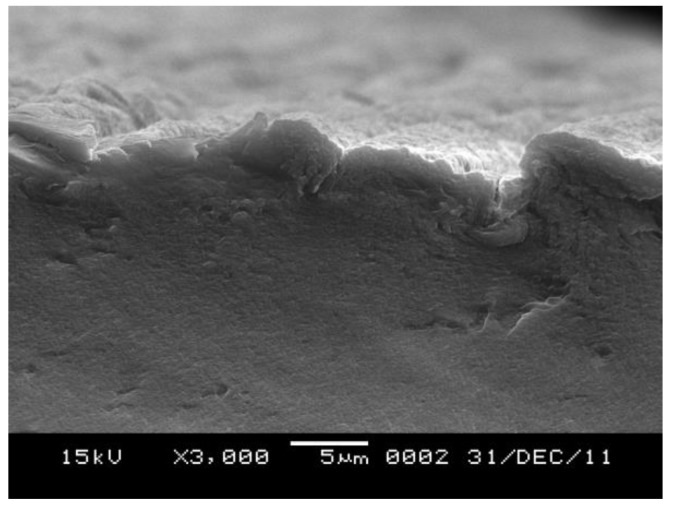
SEM of cross section of the regenerated luffa film.

**Table 1 polymers-11-00037-t001:** Experiments to determine radicals produced by plasma in water.

No.	0.1% CV (mL)	Water (mL)	Plasma Treated Water (mL)	*n*-Butanol (mL)
a	2	2	0	0
b	2	0	2	0
c	2	0	1.7	0.3

**Table 2 polymers-11-00037-t002:** Absorbance of guaiacol solutions.

NO.	a	b	c	D	e	f
Guaiacol solutions (mg/L)	95	80	65	40	35	15
Absorbance at 274 nm	1.581	1.399	1.132	0.711	0.598	0.300

**Table 3 polymers-11-00037-t003:** Degradation rate of plasma-treated guaiacol solutions.

No.	Samples	Time (min)	Absorbance at 274 nm	Degradation Rate (%)
a	guaiacol	0	2.182	/
b	guaiacol	5	1.235	44
c	guaiacol	10	0.774	66
d	luffa	20	0.710	69
e	guaiacol	15	0.652	/
f	guaiacol	20	0.485	80

**Table 4 polymers-11-00037-t004:** Main components of untreated and plasma-treated luffa fibers.

	Hemicellulose	Lignin	Cellulose
Untreated (%)	20.1	14.5	65.4
Plasma treated (%)	10.7	8.1	81.2

**Table 5 polymers-11-00037-t005:** Peak positions of untreated and regenerated luffa cellulose film.

Wavenumbers (cm^−1^)		Assignments
a	B	
3360	3360	OH stretching
2900	2900	Saturated C–H stretching
1750	1750	C=O stretching of acetyl or carboxylic acid
1646	1646	OH deforming
1500	-	Aromatic bending C–H (lignin)
1469	-	Aromatic methyl group stretching(lignin)
1429	1420	CH_2_scissoring (cellulose)
1062	1055	C–OR stretching (cellulose)

**Table 6 polymers-11-00037-t006:** Contact angle of the regenerated luffa cellulose film.

No.	Contact Angle (°)
1	16.45
2	15.28
3	17.42
The average value	16.38

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
