# Peer review of "Luffa Pretreated by Plasma Oxidation and Acidity to Be Used as Cellulose Films"

_polymers, 2018, doi:10.3390/polym11010037_

Reviewer 1 Report

The paper entitled “Luffa Pretreated by Plasma Oxidation and Acidity to Be Used as Cellulose Films” the authors described the extraction and the characterization of cellulose material from natural luffa by a pretreatment based on the oxidation and acidity of glow discharge plasma in water.The paper is interesting, but some clarifications need to increase the value of the manuscript.                                                                              

Specific Comments:               

Abstract: The authors are invited to explain in the abstract section the final application of the extracted cellulose.                                                    

Introduction-paragraph 1: the author are invited to explain better the novelty focusing the attention to the final application of produced cellulose. In this form the introduction section it is not clear.                                             

figures: the authors are invited to pay attention ate the number of the figure into the text and as caption.  The SEM images are not representative of the morphological structure of the luffa (Figure 3).

The authors are invited to insert the fractured surface of the fibres. In addition the authors are invited to comment better the figure inserting the dimension analysis of the fiber diameters.                                     

Figure7: the SEM images are not representative please change the images. The authors also invited the comment of this figure. The authors are also invited to insert a scheme of the treatment process applied and the image of raw luffa and the image of obtained cellulose extracted from luffa after the different treatments.

Author Response

Thanks for your suggestions. This is my answers.

1.      Abstract: The authors are invited to explain in the abstract section the final application of the extracted cellulose. 

The discussion about the final application of the extracted cellulose is added in the abstract, in lines 35-36 of the manuscript with “trace changes” or in lines 30-31 of the final manuscript.

2.      Introduction-paragraph 1: the authors are invited to explain better the novelty focusing the attention to the final application of produced cellulose. In this form the introduction section it is not clear.   

The discussion about the final application of the produced cellulose is added in the introduction, in lines 73-79 of the manuscript with “trace changes” or in lines 69-74 of the final manuscript.

3.      figures: the authors are invited to pay attention ate the number of the figure into the text and as caption.  The SEM images are not representative of the morphological structure of the luffa (Figure 3). The authors are invited to insert the fractured surface of the fibres. In addition the authors are invited to comment better the figure inserting the dimension analysis of the fiber diameters.   

The SEM images of the morphological structure of the luffa are changed and the fractured surface of the fibres is added. The discussion is also added. They are in the line 361-394 of the manuscript with “trace changes” or in lines 293-322 of the final manuscript.

4.        Figure7: the SEM images are not representative please change the images. The authors also invited the comment of this figure.                                                                                  

The fractured SEM image of luffa film is changed. It is in the lines 465-472 of the manuscript with “trace changes” or in lines 388-395 of the final manuscript.

5.      The authors are also invited to insert a scheme of the treatment process applied and the image of raw luffa and the image of obtained cellulose extracted from luffa after the different treatments.

Yes, it is added in lines 573-574 of the manuscript with “trace changes” in lines467-468 of the final manuscript.

6.      Other modifications are marked in the manuscript with “trace changes”.

Reviewer 2 Report

The manuscript entitled: "Luffa Pretreated by Plasma Oxidation and Acidity to Be Used as Cellulose Films" is an interesting work dealing with a new environmentally friendly way for the production - extraction fo cellulose out of natural resources. This is interestign since the environmental way of the production can be considered to save resources of our planet. However, there are a few comments that I would like that the authors answer and also clarify in their manuscript: 

1. To what extent are the properties of the produced cellulose the same like the commercial or the prepared by other ways cellulose? Can the authors provide some comparison tests?

2. Is it indeed more environmentally friendly to extract cellulose out of luffa? How much energy is consumed for the Plasma? I assume that it is not that high, but I would like that the authors make a short and concrete statement of the cost of the production as well as to the efficiency of the proposed method.

3. Regarding characterization: For the SEM the authors investigate the surface f the treated and not treated material. Do the authors also check the cross-section of the materials to investigate the inner surface?

Author Response

Thanks for your suggestions. This is my answers.

1.       To what extent are the properties of the produced cellulose the same like the commercial or the prepared by other ways cellulose? Can the authors provide some comparison tests?

I provide some comparisons. It is added in lines 334-339 of the manuscript with “trace changes” or in lines 266-271 of the final manuscript.

2.       Is it indeed more environmentally friendly to extract cellulose out of luffa? How much energy is consumed for the Plasma? I assume that it is not that high, but I would like that the authors make a short and concrete statement of the cost of the production as well as to the efficiency of the proposed method.

The discussion about the energy consuming of plasma treatment is add in lines 351-360 o f the manuscript with “trace changes” or in lines 283-292 of the final manuscript

3.  Regarding characterization: For the SEM the authors investigate the surface f the treated and not treated material. Do the authors also check the cross-section of the materials to investigate the inner surface?

The SEM images of the morphological structure of the luffa are changed and the fractured surface of the fibres is added. The discussion is also added. They are in the line 361-394 of the manuscript with “trace changes” or in lines 293-322 of the final manuscript.

4  Other modifications are marked in the manuscript with “trace changes”  .

Reviewer 3 Report

The author should revise the title to make it more comprehensive.

The method for standard curve of lignosulfonate was described in the experimental section, but no experimental data was provided in the results section, only the maximum absorption peak was mentioned by text. Thus, the result of orthogonal experimental is questionable.

About the XRD analysis, the citation 11 was mentioned to explain the change in luffa structure. However, this citation is a domestic publication which is difficult for reviewer to access and check the validity of the result. Similar to the citation 13, which was mentioned in the introduction.

The FTIR figure is not clear and the interpretation of FITR results is not convincing. For instance, in figure 6, the peak at 1750 cm-1 appeared in sample (b), but was not mentioned in table 5. The 2 peaks at 1469 cm-1   and 1500 cm-1 in sample (b) might be overlapped with other peaks, instead of disappeared. The authors should provide a zoomed-in figure on this region to clarify this result. 

Comparison between the mechanical properties of plasma pretreated film and untreated film should be conducted, in order to address the effective of plasma treatment.

Author Response

Thanks for your suggestions. This is my answers.

1.The method for standard curve of lignosulfonate was described in the experimental section, but no experimental data was provided in the results section, only the maximum absorption peak was mentioned by text. Thus, the result of orthogonal experimental is questionable.

I am sorry for forgetting the recording the data of the whole curve. So there are only the absorbency peaks in the original manuscript. To answer this question, I use the experimental data of guaiacol which is used as the model compound of lignin. The experimental of guaiacol was done at the same time with lignosulfonate.

So the experiment part is modified. It is in lines 140-156 of the manuscript with “trace changes” or in lines 116-132 of the final manuscript.

The discussion part is also modified. It is in lines in 289-360 of the manuscript with “trace changes” or in lines 221-293 of the final manuscript.

2.About the XRD analysis, the citation 11 was mentioned to explain the change in luffa structure. However, this citation is a domestic publication which is difficult for reviewer to access and check the validity of the result. Similar to the citation 13, which was mentioned in the introduction.

In the XRD, the discussion about the citation 11, the lines “434-436” of the manuscript with “trace changes” is deleted. The citation 13 is changed to an English citation of 13 in line 554 of the manuscript with “trace changes” or line 469 of the final manuscript.

3.The FTIR figure is not clear and the interpretation of FITR results is not convincing. For instance, in figure 6, the peak at 1750 cm-1 appeared in sample (b), but was not mentioned in table 5. The 2 peaks at 1469 cm-1   and 1500 cm-1 in sample (b) might be overlapped with other peaks, instead of disappeared. The authors should provide a zoomed-in figure on this region to clarify this result.

The explain about the at 1750 cm-1 appeared in sample (b) is added in lines 452-455 of the manuscript with “trace changes” or in lines 378-382 of the final manuscript. 

About peaks at 1469 cm-1 and 1500 cm-1 in sample (b), I provide a zoomed-in figure of FTIR to clarify the result, which is not overlapped with other peak.  

4.Comparison between the mechanical properties of plasma pretreated film and untreated film should be conducted, in order to address the effective of plasma treatment.

I did experiment again. The untreated lufa was coated with much lignin, which cannot dissolute in the solvent. So the fim cannot be prepared. 

5.Other modifications are marked in the manuscript with “trace changes”.

Round  2

Reviewer 1 Report

The authors have modified the manuscript as suggested. I have no hesitation for the publication.